# Researching Teacher Work Motivation in Ghana through the Lens of COVID-19

**Michael Agyemang Adarkwah**

Smart Learning Institute, Beijing Normal University, Beijing 100875, China; adarkwahmichael@bnu.edu.cn

**Abstract:** Teachers, particularly in developing contexts, were vulnerable populations during the COVID-19 pandemic. As natural parental figures for students, they had to reconcile the dual role of ensuring the safety and health of students and their own and their family's well-being. The external crisis of COVID-19 heightened the negative experiences of teachers in their work environments during both online and physical instruction. This qualitative phenomenological study involving thirty (30) secondary school teachers in Ghana took a comprehensive and fresh look at how COVID-19 impacted the work motivation of teachers. It was found that teachers suffered a great deal of stress in the wake of the pandemic and had to face mounting concerns about their working conditions. The low morale of teachers precipitated by COVID-19 made them develop attrition intentions. However, intrinsic and altruistic traits such as passion, the feeling of responsibility, and the desire to contribute to society and foster student development made teachers resilient towards the deleterious effects of the pandemic to promote optimal teaching. Future studies should investigate the installation of support structures that strengthen the motivation of teachers in unforeseen crises.

**Keywords:** teacher motivation; work motivation; job satisfaction; COVID-19; Ghana

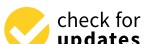



## 1. Introduction

The exacerbated tension and stress from the COVID-19 pandemic had the potential to negatively affect the work motivation of teachers. The unpreparedness of teachers for the challenging times prompted an inquiry into whether COVID-19 strengthened or weakened the work motivation of teachers. While teachers might need time and space to process and ensure their own well-being in these times of pandemic, they are naturally parental figures for their students [1]. In some situations, reconciling these two roles can add stress on teachers, having a negative impact on their motivation to teach. The internal crisis of teacher stress that causes them to become fatigued by the profession was worsened by the external crisis of COVID-19 [2].

Since March 11, 2020, when the World Health Organization (WHO) avowed COVID-19 as a pandemic [3,4], teachers have found themselves in a difficult situation regarding how to instruct students [5–7], ensure the welfare of their families [1], deal with financial constraints from the pandemic [8], and how to mitigate the psychological impact the pandemic can have on them [5,9]. Arguably, it can be said that COVID-19 made the work of teachers risky [3]. Teachers need to be supported in multiple ways for them to be effective in unpredictable circumstances such as the COVID-19 pandemic [8]. There will be no complete educational recovery without empowered, motivated, and effective teachers [10].

Westphal et al. [11] and Robinson et al. [12] reported that during the COVID-19 pandemic, there was much focus on the situation of students with less or no focus on teachers such as how the pandemic has impacted them and the need to address issues relating to their working conditions. In agreement, Trinidad [2] added that almost all studies focused on the impact of the pandemic on the academic performance, learning outcomes, and the physical or mental health of students. For example, Sulaiman et al. [13] mentioned that in the educational arena, the well-being of students was of the utmost

importance during the COVID-19 pandemic and called for teachers to understand the needs of students. A desk review of major scientific databases such as Web of Science (WOS) and Scopus, and academic search engines such as Google Scholar revealed that prior investigation on the COVID-19 impact on teacher work motivation took a narrow focus on either their motivation to engage in online learning [6,14], professional training to use digital tools for instruction [15,16], or the psychological impact of the pandemic [5,9]. To date, the effect of the crisis period on the overall work motivation of teachers from the global south has not been perfectly understood, particularly in African countries, where the work motivation of teachers was a problem pre-pandemic.

It is essential to take a fresh and comprehensive look at teacher work motivation in light of the COVID-19 pandemic. Especially in the sub-Saharan African region, little is known about how teachers perceived their motivation during the pandemic, the support systems available for teachers in the distressing times, their retention and attrition intentions, and how they are navigating pedagogical instruction in the "new normal." The work motivation of teachers is of much importance because it is linked to their long-term commitment to teaching, quality of teaching, and mental health [14]. Fostering teacher resilience against the spillover effects of COVID-19 by ensuring they are motivated will be vital for education recovery and transformation [8]. The COVID-19 pandemic offers a suitable context to explore how motivated they are to teach in adverse times and the kind of support they hope to receive.

This study's findings highlight issues relating to the work motivation of teachers for post-COVID education recovery and transformation and can serve as a blueprint for school leaders and administrators, policymakers in education, and relevant school stakeholders to appropriately tackle issues relating to teacher dispositions at work to promote educational success. The overarching question guiding this explorative study is, "how do teachers perceive their work motivation during the initial wave of the COVID-19 pandemic?" Specifically:

RQ1. To what extent did COVID-19 affect intrinsic, altruistic, and extrinsic motives to teach?

RQ2. What are teachers' perceptions of governmental and institutional support to enhance their motivation during the COVID-19 pandemic?

RQ3. How has COVID-19 affected the retention and attrition intentions of teachers?

RQ4. How are teachers navigating pedagogical pathways in the pandemic crisis?

## 2. Study Background

Prior studies have revealed that teachers in low-income countries are not adequately motivated in their teaching practice [17,18]. According to the authors, the demotivation of teachers in developing countries serves as one of the principal reasons why teaching practice and student learning have deteriorated over the years in Africa. Wolf [19] adds that teachers in low-income countries are not sufficiently equipped for the difficulties they encounter in teaching and face many hurdles in their personal life as well as work, which hinders classroom effectiveness and their general well-being. Ghana, as with many other African countries, faces issues of teacher attrition, teacher strikes, and teacher dissatisfaction, which are connected to teacher work motivation [20–26].

A report published by the Teacher and Educational Worker Union and The Ghana National Association of Teachers (GNAT) revealed that 10,000 teachers in the Ghana Education Service (GES) abandon their posts to pursue other adventures for diverse reasons annually [27]. In 2013 alone, Mrs Irene Duncan-Adanusa, who is the secretary of GNAT, announced that an estimated amount of 33,185 teachers in Ghana abandoned their posts for greener pastures, and this has affected the learning outcomes of students [28]. Affull-Broni [29] concluded that the motivation of teachers in Ghana, especially in higher education, is weak.

Several policies enacted by the government to promote education in Ghana have excluded teacher work motivation, which is key to improving the academic performance

of students [30,31]. Osei (2006) mentioned that although the government of Ghana expects much from teachers, it does not adequately reward them. Newspaper reports have revealed that teachers in Ghana have consistently embarked on strike actions because they feel they are not well-motivated [32,33]. On August 19, 2019, Mr Eric Agbe-Carbonu, President of the National Associate of Graduate Teachers (NAGRAT), announced that teachers have embarked on strike action because government policies do not factor in teacher work motivation [26,34]. Specifically, the teachers lamented that concerns about their welfare were not resolved, including: the non-payment of salary arrears, delays in promotion, the imposition of an insurance policy on teachers, and the appointment of non-GES personnel to manage affairs at the GES [35].

The Ministry of Education policy [36] declared their acknowledgement of teachers at all educational levels as critical figures in the attainment of quality education, and that all promulgated policies on education would have the welfare of teachers at heart in succeeding years ahead. The policy further stated that they would provide teachers with incentives, allot money to pay all their arrears, furnish them with cost-effective accommodation, grant them myriad opportunities for them to develop professionally, restore the allowances of trainees, and many more. However, the consistent threat of teachers going on strike and the multiplicity of complaints that their welfare is not taken into consideration serve as evidence that factors affecting the motivation of teachers are critical issues in the country, and this has subsequently affected student learning in the country.

By way of explanation, teachers in Ghana have constantly lamented about the work motivation pre-pandemic. With the sudden surfacing of the pandemic, the challenges of teachers in the country have intensified, meriting the need to take a new and thorough look at the work motivation of teachers in the country. That is, it is important to explore how the pandemic affected teachers as they transitioned to new modalities of instruction, experienced varying levels of stress, and risked their health if they taught in person [2]. Teacher work motivation is central to the retention and attrition intentions of teachers. Studies carried out in the United Kingdom and Israel revealed that COVID-19 is likely to increase the attrition rates of teachers [4,37]. To attract and keep quality teachers in the labour force to aid in mitigating the educational loss from the pandemic, teachers need to stay motivated, especially in Ghana, which has a track record of high teacher turnover and attrition [27,28].

The first COVID-19 case in Ghana was recorded on 12 March 2020, just a day after WHO declared the novel coronavirus as a pandemic. The COVID-19 health restrictions and other sweeping measures adopted by the Ghanaian government to curb the spread of the virus greatly impacted 9.2 million learners from kindergarten (KG) to SHS, and 500,000 tertiary students [38]. Pre-pandemic education in Ghana was more "analogue" in nature and less "digital." That is, teachers and students were accustomed to the old-fashioned way of education (the "chalk and talk" approach where teachers speak and students listen in constructed classrooms). The lockdown measures, which caused an abrupt disruption in education, triggered an outbreak of online learning among secondary and post-secondary institutions in the country. However, because there was high resistance to online education in the past, the emergent adoption of online learning was accompanied by a myriad of challenges that frustrated both teachers and students.

For example, interview data from a recent study suggest that the experiences of teachers during the pandemic were negative due to environmental constraints and inadequate institutional support [39]. Teachers at the university level who did not have prior virtual learning experiences also met with difficulties in instructional delivery. In the study by Salifu and Abonyi [39], the two techniques teachers used in managing the challenge involved regulating the behaviour of learners and controlling instructional content. In one study, a cohort of preschool teachers also enumerated several challenges including increased workload, emotional drain, and financial difficulties [40]. The educational quality in Ghana was already low before the pandemic, but the challenges that emerged in the aftermath of the COVID-19 crisis compounded the vulnerability of teachers and school children [19].

After the sudden closure of schools and the experimental online learning in 2020, there was a temporal resumption of face-to-face learning for final years who were preparing to write the West African Senior School Certificate Examination (WASSCE) before a nationwide resumption of school operation [38,41]. Teachers who were exhausted because of the "challenge-ridden" online learning had to show up physically on the job to instruct students during the fatal phase of the pandemic. While the choice to resume physical learning may have eased parents' burden and benefited students, it posed a threat to the health and safety of teachers [42]. Ascertaining how to boost the morale of such teachers is essential for quality education recovery.

## 3. Literature Review

### 3.1. Teacher Work Motivation

Motivation is conceptualized as the underlying reason for a particular behaviour [43]. Gagne [44] defined the work motivation of teachers as the inclination to instruct and a person's relational attitude toward students during classroom instruction. Wang et al. [45] also defined teacher work motivation as "a stable psychological tendency to meet the needs of teachers and promote teachers to complete the specific goal of teaching and educating people" (p. 253).

In this study, teacher work motivation is operationalized to mean a teacher's passion for instructing students for the students to engage in meaningful learning even in unfavourable working conditions. That is, teacher work motivation has to do with all internal and external factors that give teachers the morale and enthusiasm to ensure quality teaching for school progress.

There is concrete evidence that teacher work motivation is influenced by educational reforms, and the work motivation of teachers subsequently exerts an impact on the learning outcomes of students [46]. Teachers should possess some level of enthusiasm to be able to overcome the challenges pertaining to classroom instruction and improve students' learning outcomes [47]. For major school improvement, teacher motivation has been identified by researchers as an important factor [48]. Davidson [49] concurs that the work motivation of teachers plays a key role in their level of commitment and how they study and engage in the classroom. When teachers are supported through motivational strategies, it promotes their welfare and facilitates effective learning in schools [50].

Thus, among instructors, those who often perform better than their colleagues are those who are well-motivated [51]. Klassen et al. [52] stated that across different contexts and diverse cultures, the work motivation of teachers is often associated with increased teacher commitment, high quality of teaching, and classroom engagement that affect the learning outcomes of students. Teachers who are likely to succeed in their jobs are those who are highly motivated [53]. That is, comparatively, motivated teachers perform higher than demotivated teachers [54]. It can be said that the flip side of motivation is demotivation. While teacher work motivation helps both teachers and students to realize better outcomes, their demotivation leads to negative outcomes for both parties [55].

### 3.2. Types of Teacher Work Motivation

3.2.1. Intrinsic Motivation

Intrinsic motivation simply means inherently doing something because it brings joy and is interesting [56]. According to the authors, it is an inherent or natural tendency to extend one's abilities to explore and seek out novelty and difficulties. Thus, naturally, one is inclined towards spontaneous interest, mastery, assimilation, and exploration, which play a key role in an individual's social and cognitive development. Activities that are intrinsically motivated are not dependent on external pressure or incentives, instead, they give an individual satisfaction or joy [57].

Intrinsic motivation serves as a determinant of the degree of an individual's willingness to engage in something for the person's sake rather than external rewards [58]. Individuals who are intrinsically motivated find joy in carrying out a task and tend to invest

more effort in undertaking the task [59]. High task performance is associated with intrinsic motivation [60]. Intrinsic motivation has also been associated with teaching education and teaching such as the reason for selecting teaching as a professional job, engagement in pedagogical activities, and deep learning [61].

Teachers' instructional practices through their achievement goals are affected by intrinsic motivation [62]. It has also been found that a positive correlation exists between intrinsic motivation and teachers' general pedagogical knowledge (GPK) and GPK change [63], and results in less burnout, and more career optimism in novice teachers [64]. Ryan and Deci [57] revealed that intrinsic motivation is an inner resource that can decline if schools do not create need-supportive contexts (such as a supportive climate that is a catalyst for autonomous motivation) that ensure psychological need satisfaction.

### 3.2.2. Altruistic Motivation Factors

Altruism refers to "the willingness to incur a loss of material welfare to enhance the welfare (material or not) of others" ([65], p. 186). Altruistic motivation is a particular type of motivation whereby the ultimate goal is to increase the welfare of others [37,66]. Li et al. [67] add that altruism is a kind of behaviour that is beneficial to people at a personal cost. Thus, individuals who demonstrate altruism help others and provide support for ideas or concepts even though extrinsic rewards may not be guaranteed [68]. Altruistic behaviour can be demonstrated in the form of money, time, or physical sacrifice to enhance the well-being of others [69] and contribute to society [70]. Antecedents of altruistic motivation are empathy, affording support, active protection of those who are helpless, experiencing shame, and having a fear of victimization [65,71,72].

Altruistic individuals have the pure desire to help others, and an urge to sacrifice for the good of others without expecting rewards or compensation [73,74]. Fray and Gore [75] mentioned that altruistic reasons are one of the most important influences causing people to enter the teaching profession. Altruistic motivation such as "serving the next generation" was found to be an underlying reason for pre-service teachers' decision to select teaching as their professional work [61]. Teaching is a profession that demands a high degree of altruism [76]. A survey of practising teachers in Australia revealed that altruistic motivation such as contributing to children and adolescents was one of the reasons why people enter the teaching profession [77].

### 3.2.3. Extrinsic Motivation

Extrinsic motivation is defined as undertaking a course of action to obtain a separable outcome [56,57,78]. For extrinsic motivation, the force that drives an individual to manifest a behaviour is derived from an external source [79]. Extrinsic motivation is characterized by instrumental gain and loss, such as incentives [80]. Extrinsic motivation is driven by incentives and rewards that are considered as controlling or pressuring a person to act in a particular way [80]. Some of the common extrinsic motivation factors affecting the work motivation of teachers include financial incentives, opportunities for promotion and professional development, school environment, etc. According to the self-determination theory (SDT) framework by Ryan and Deci [56], extrinsic motivation factors can have an impact on the intrinsic motivation of an individual. COVID-19 in this study is an external factor that has the potential to alter or undermine the intrinsic and altruistic motives of a teacher.

## 4. COVID-19 and Teacher Work Motivation

As reviewed earlier, there are a few studies that have taken a comprehensive look at the work motivation of teachers in light of the pandemic. Nonetheless, teacher work motivation research that focuses on aspects of education such as online learning, teacher professional development in using digital tools for technology-enhanced learning, and coping strategies of teachers during the pandemic all come to a consensus that COVID-19 gravely threatened the work motivation of teachers [5,6,14,15].

For example, there was faculty resistance to embracing online learning, financial constraints in procuring digital tools for instruction and purchasing data bundles, difficulty in using new technologies to teach, and a lack of professional orientation on new methods of instruction, which all affected the work motivation of teachers [13,41]. The psychological impact and social uncertainties of the pandemic, which manifested themselves in the form of anxiety, stress, depression, protests, and ambiguous and conflicting roles, also compromised the work motivation of teachers [1,9].

According to Westphal et al. [11] COVID-19 crisis affected the behaviour of teachers and their retention. Collie [81] links teachers' somatic burden, stress, and emotional exhaustion during the COVID-19 pandemic to poor working conditions in the school environment and leadership style. With regards to school leadership, Trinidad [2] expounded that organizational factors such as mentorship and organizational decisions such as choosing the modality of instruction were consequential for teachers' feeling of satisfaction to rapidly engage in teaching. That is, organizational support systems predicted personal satisfaction or burnout in teachers.

Chiu et al. [6] mentioned that in this pandemic era, teacher well-being is often perceived as "nice to have" but not "essential" and is often dominated by negative mental health. Overlooking the well-being of teachers and the difficulties they encounter will lower their morale and ultimately curtail the desire to promote quality post-pandemic education.

## 5. Method

### 5.1. Research Design

This research was carved from a larger study about how job satisfaction mediates the impact of teacher work motivation on the academic performance of students. In this present study, a qualitative design was employed. Specifically, the phenomenological design was adopted to understand the lived experiences of the teachers regarding their working conditions during the COVID-19 pandemic. Unlike just any qualitative approach, phenomenology helps the researcher to "see the world" Adams and van Manen [20] investigate the natural attitudes of participants in the real world [82]. Langdridge [83] defines phenomenology as "a method that hopes to throw light on the perspectives of individuals regarding the world in which they find themselves and what those perceptions signify to them" (p. 4). Interpretive phenomenology, also known as hermeneutics and advocated by Heidegger, was used. Interpretive phenomenology focuses on describing, understanding, and interpreting individual experiences.

### 5.2. Participants, Sampling, and Data Collection

Purposive sampling, which simply refers to making a judgement on the best participants to provide answers to the research questions to achieve the aim of the study was used to recruit thirty (30) teachers from thirty (30) "Category B" secondary schools in the Ashanti Region. The dozen secondary schools in Ghana are categorized as "A", "B", "C" and so on in order of rank. The selected teachers participated in COVID-19-induced online learning and face-to-face (F2F) learning during the pandemic. As of December 2022, although COVID-19 is not totally obliterated in Ghana, these teachers were engaged in physical learning because of the challenges of the online mode. The teachers in the study taught core subjects; English ($n = 10$), mathematics ($n = 4$), social studies ($n = 7$), and integrated science ($n = 9$) and had a teaching experience of at least three years. Overall, there were nineteen (19) males and eleven (11) females. The age of the teachers was: 25–30 (13.3%), 31–36 (30%), 37–42 (36.7%), 43–48 (10%), 49–54 (6.7%), and 55–60 (3.3%).

Participants were informed about the aims of the study and were assured that neither they nor their host institution would incur any risk by their voluntary participation. Informed consent was gained before the interview session began. The semi-structured interview sections occurred in the school environment of each of the participants at the time of their convenience and lasted for approximately fifty (50) minutes for each of the participants. There were rounds of breaks for participants when necessary. All the inter-

views were conducted in English and were audiotaped. The audio files were transcribed verbatim and stored in a secure location with no identifying details.

*5.3. Instrument*

The main instrument for the study was a self-designed interview guide comprising a list of semi-structured questions. The total number of items on the instrument was twenty-one (21). The first section of the interview guide was made up of demographic-related questions and the second section contained the body of questions. The development of the instrument was underpinned by literature. Several relevant research studies related to the current research were reviewed to guide the researcher in formulating the research questions on the instrument [4,5,9,27,37,41,52,80]. To enhance the validity of the instrument, a peer debriefing approach was used. All suggestions requested were made. Some of the questions in the final draft of the interview guide included; "How do you perceive the online instruction during COVID-19?" "What major challenges did you face during the instruction?" "Tell me some of the main psychological stressors you experienced with teaching," "What kind of school resources were available to assist you in your job during the pandemic?" "What were the main support systems for teachers during the pandemic?" "What impact has COVID-19 and its after-effects had on your turnover and attrition intentions?" "How you do navigate through the challenges?", "What are the main reasons why you endured the challenges and instructed students?" (see Appendix A).

*5.4. Ethical Issues*

The research was fully approved by the Ashanti Regional Education Office through its ethical review process. A letter constituting the objectives and the instrument of the study was submitted for examination before approval. The Education secretariat was assured that no school would be named in the research. Additionally, a letter of introduction was submitted to the principals of the schools where the teachers were recruited from. Only teachers willing to freely respond to the interview questions were signed up for the study and were informed they could withdraw at any stage of the data collection. Moreover, to avoid the tracing back of data to the participants, they were informed pseudonyms would be used in place of their names. Because data were collected on-site during the peak days of the pandemic, all advocated COVID-19 health protocols were observed.

*5.5. Data Analysis*

Braun and Clarke's [84] systematic guide on thematic analysis was followed to conduct the qualitative analysis. The written form of the interviews was read repeatedly by the researcher to ensure familiarity with the data collected and understand the depth and breadth of its contents. Participants in the study were given the liberty to access the transcribed versions of their responses to ensure the accuracy of data. After making meanings of the content of the interviews, codes were manually generated and they were further grouped into different themes and sub-themes. All the themes were reviewed and refined to avoid the wrong interpretation of data. Additionally, an independent researcher assessed the generated codes for validation. Codes and themes that were not coherent with data were collapsed. The main themes were defined and exemplars were located as evidence of participants' emotions, experiences, and perceptions regarding their ordeal (work motivation) during the COVID-19-engineered online education and F2F learning.

## 6. Results and Discussion

The themes and sub-themes generated were put into four (4) categories created to reflect each of the research questions (Table 1). Overall, twelve (12) main themes were identified. From the participants' responses, there were key issues that resulted in low work motivation: the challenges of online learning in the early phase of the pandemic, the social discrimination associated with them physically showing up at their jobs, psychological stressors of the pandemic, inadequate school resources, and a new policy known

as the Multitrack Year-Round Education (MTYRE). The principal support systems were the government and their institutions. However, their experiences with receiving support were sometimes negative. As a result of the negative experiences in the aftermath of the COVID-19 pandemic, teachers in the study had intentions to leave the profession in pursuit of white-collar jobs and further studies. Nonetheless, six participants in the study mentioned that they made the decision to remain in the profession because they were nearly due for promotion. The intrinsic–altruistic motives of teachers propelled them to teach in the challenging times. It is inferred from the findings that traits such as passion, feelings of self-respect and a responsibility towards students, a desire to contribute to society and student development can encourage a teacher to persevere during unprecedented difficulties such as COVID-19 and boost their morale to continue teaching students. Moreover, such unforeseen events can trigger attrition and turnover intentions.

**Table 1.** Categories, Themes, and Sub-themes.

| Categories | Themes | | Sub-Themes |
|---|---|---|---|
| RQ1. Antecedents of demotivation | 1. | Online learning | Lack of ICT tools, network lag, no data bundle. |
| | 2. | F2F COVID-19 discrimination | Societal discrimination and low professional identity |
| | 3. | Psychological stress | Anxiety about physical safety and family health, Dealing with student safety |
| | 4. | School resources | Lack of facilities (e.g., sickbay) and teaching materials. |
| | 5. | MTYRE-induced burnout | Lack of holidays, pressure |
| RQ2. Support systems | 6. | Government support | Education policies, PPEs |
| | 7. | Institutional support | Professional development, motivational package. |
| RQ3. Retention and attrition intentions | 8. | White-collar jobs | Banking sector, social work, secretary of firms. |
| | 9. | Further studies | Apply for master's |
| | 10. | Promotion | Stay in the job for upgrade |
| RQ4. Navigating pedagogical pathways | 11. | Intrinsic motivation | Passion, feelings of responsibility and self-respect |
| | 12. | Altruistic motivation | Contribution to society and helping students |

*6.1. Antecedents of Demotivation*

6.1.1. Theme 1: Online learning

The pandemic-born online learning in 2020 was accompanied by many challenges that teachers in the study perceived as negatively affecting their motivation to teach. In Ghana, online education was yet to gain roots and was rarely used in secondary education. As a result, the emergent adoption of online learning as a panacea to the disruption caused by COVID-19 was not warmly welcomed by the teachers. Teachers were faced with poor internet connectivity, no WIFI and data bundles, and the lack of digital tools such as a

computer to facilitate the instruction process. Schools that were privileged to have ICT laboratories sometimes did not have or had limited functioning computers. Almost all the participants expressed that they had low morale because of these challenges.

> "In my school, we are not used to online education. When there were school closures, we attempted to try that but it didn't work until we were cleared to physically return to school. You know, the online education can be frustrating. My school did not provide teachers and students with the tools to use and we did not have any WIFI. Our ICT lab is also not in good shape"—John

Online learning in Ghana during COVID-19 was unsuccessful because of situational challenges that curtailed smooth instruction delivery [63,85,86]. The main online learning that occurred in senior high schools was the creation of a virtual platform by the Ghana Education Service (GES) for students, which included an online (icampus) program and the provision of educational content via radio and television programs. Some schools privileged to have ICT laboratories also made use of them for instructional delivery, although there were accompanied challenges. As a result, there were fewer teacher–student interactions. Prior studies that investigated online learning in pre-tertiary institutions in Ghana all came to the conclusion that COVID-19-induced online learning failed [87,88]. Teachers preferred F2F learning as opposed to the online modality of instruction. That is, in many second-cycle institutions, online learning was halted because of the challenges experienced by both teachers and students.

### 6.1.2. Theme 2: F2F COVID discrimination

Some of the teachers also indicated that the public had a negative image of them already that was worsened by the pandemic. The teachers regarded teaching as a profession that was highly respected but had gradually lost its respect over the years because of the treatment of teachers. It was mentioned that during the pandemic, when teachers physically went to work when there were lockdown measures in place, some neighbours perceived them as possibly having the virus and were hesitant to at times get close to them. This was perceived as a form of discrimination by the public compounding it with the existing low professional identity in the country.

> "Sometimes, when you return from school, people think you have the virus because you went out. They are afraid to even come close to you. For me, I live in a compound house so you can imagine. I used to feel sad because of that but it was not a big problem for me because I knew I did not have the virus. I come home to rest and then go back to teach"—Benjamin

During the COVID-19 pandemic, many social groups suffered from stigma and discrimination because they were perceived as carriers of the novel coronavirus or posed a threat to effective social living [89]. In Ghana, teachers and health workers who physically showed up at work were perceived to be a potential source of infection and were at times shunned by their immediate communities [3].

### 6.1.3. Theme 3: Psychological stress

A majority of the teachers opined that the COVID-19 pandemic brought a lot of stressors such as the thought of contracting the virus or a loved one dying from it. Additionally, the teachers were responsible for the well-being of their students at school. This was an added burden that affected the psychological health of students. One teacher added that although he heard there were psychological support services for teachers, he never experienced it in his school.

> "Because even the health experts were unsure about the nature of the virus, it brought a lot of anxiety. You always see the death toll and the number of infections on television. I was afraid for myself and my family. The government wanted us to go to school to teach in person so it was a concern for me. The teachers also had to take care of students at school"—Samuel

"No, I wasn't provided with any psychological counselling"—Angela

The deleterious effect of COVID-19 had the potential to extend from physical injury to substantial psychological consequences such as anxiety, depression, panic, suicidal ideation, and delirium [5]. It has been underscored by prior studies that the mental health of teachers was greatly impacted by COVID-19 [5,9,42]. Teachers experienced emotional distress and paranoia because of the pandemic.

### 6.1.4. Theme 4 School resources

According to some of the teachers, during the COVID-19 pandemic, there was limited accommodation for students on campus and sickbays to house them in case any of them became infected. Additionally, facilities such as ICT centres that could help them in their teaching were limited. There was also a delay in the supply of teaching and learning materials. This made the working conditions at school difficult at times for some of the teachers.

"In my school, we had a sick bay but there were not enough beds for all the students who were sick. Some of their peers were afraid a friend might have the virus if they were sick. So teachers had to deal with the situation and calm things down. When any problem arises, it comes back to teachers. You have to always maintain order and make sure no one is afraid. I think the school needs to build more sickbays or expand the current one"—Lucy

"Availability of teaching materials in Ghana is very low. At times you have to buy the teaching materials yourself. Maybe, the COVID-19 affected the distribution of funds or resources to the schools. I'm not really sure. But this has always been the case"—Bright

Past studies in Ghana have highlighted the lack of adequate school infrastructure for effective teaching and learning [38,87]. Especially during the COVID-19 era, the lack of adequate school resources was apparent as schools grappled with integrating ICT into education and providing quality health services for students who might have been infected by the coronavirus. Overcrowding and infrastructure deficits in many developing countries represented a barrier to the safe reopening of schools post-COVID-19 [90]. The study by Sanni et al. [90] reported that in Nigeria, only 6.9% of schools employed quality health personnel. For institutions with sick bays, the concern was the satisfaction of use.

### 6.1.5. Theme 5: MTYRE-induced burnout

An ample number of the participants were observed to suffer from burnout as a result of the new MTYRE policy that was implemented during the physical resumption of school. In Ghana, this approach is popularly known as the "double-track" system and was adopted because of limited facilities in the secondary schools in the country. In MTYRE, students are split into two groups with the groups alternating with each other every semester in two terms. Thus, teachers provide instruction to two groups of students at different times. Due to this approach, teachers were robbed of their holidays and experienced increased pressure in the classroom because of the limited time to instruct a particular batch of students.

"I understand that the double-track was in a way helpful because we could maintain social distancing and teach both groups of teachers. But teachers suffered because of this. There was an increased workload. Immediately you finish teaching one group, the other group also returns to school and you have to teach the same thing all over again using less time. It brings pressure"—Julius

The MTYRE increased the working hours of students and also robbed teachers of their vacation. With the advent of COVID-19, the MTYRE became increasingly popular. While in the pandemic era, the MTYRE led to a reduction in class size and effective usage of teaching facilities, Adarkwah [91] reports that it led to poor learning outcomes for students

due to behavioural problems of students. The short duration of the academic semester due to the MTYRE system also led to the school curriculum not being completed.

*6.2. Support Systems*

6.2.1. Theme 6: Government support

Most of the teachers in the study perceived educational policies during COVID-19 as not having the interest of teachers at heart. Some of the teachers mentioned that their colleagues at the basic school largely stayed at home, while they were tasked to instruct final-year students. Moreover, they perceived COVID-19 policies to be more favourable to health professionals than teachers. Notwithstanding this, it was added that the government supplied schools with personal protective equipment such as gloves, Veronica buckets, and face masks to help curb the spread of infection at school.

> "Because we were preparing final year students for their exams and the pandemic set up, we were called back to school to teach. Maybe they could have found a better way to help us prepare the students instead of calling us to school all of a sudden. It endangered our lives because we were also afraid of the virus by then. Even in the past, school policies have not been favourable to teachers at all. Student discipline is even a problem because of some new policies that restrain teachers. You have to find a way for students to adhere to the COVID-19 protocols"—Martin

> "Oh, as for face masks and sanitisers, the government brought some to our school for teachers to use. But it was not enough. You had to buy some personally too. But at least, it was good on the part of the government"—Joyce

Government support for schools was minimal during the onset of the pandemic. It could be argued that more attention was paid to the health sector than the education sector [3]. As reviewed earlier, pre-pandemic, teachers perceived governmental policies as unfavourable to teachers. The experience of teachers in receiving support from the government account for one of the reasons why they are demotivated [3,22,92].

6.2.2. Theme 7: Institutional support

There were motivational packages that were variably felt by the teachers. While some of the teachers responded that they received some form of incentive, some mentioned that they did not. The incentives were in a form of a monetary allowance to motivate teachers who were engaged in physical instruction. A teacher who did not receive the motivational package indicated:

> "Even though I was not given any motivation to teach, I understand the system very well so I was willing to do my work without letting the attitude of the educational system affect my delivery of lessons in the classroom"—Elisha

> "I am disappointed with the system but I do not let this affect my work. I have equal qualifications as some of the teachers or even better but I learnt some of the teachers received some kind of motivation while I did not receive any. I used to teach the pupils during the COVID-19"—Audrey

Two teachers also added that during the early phase of the COVID-19 pandemic in 2020, they did not receive any professional training about online education but were briefly oriented about the health instructions to follow.

> "As for the online learning, everything was fast. We didn't receive any formal orientation about it. That explains why we had to stop that idea. But when we returned to school to teach in person, we had a staff meeting where we were informed about the health restrictions and guidelines to prevent a mass spread of infection at school"—Austin

> "I didn't receive any training. About the training, only one teacher and the head were trained in my school"—Francis

Because teachers were unaccustomed to digital instruction, it was necessary for them to receive some kind of technical support or training in using ICT tools for instruction during the COVID-19 pandemic. However, as reported by studies reviewed earlier, there were fewer professional development programs to ensure teachers were technically adept at instructing students online [3,41,86,87]. The perceived lack of motivational incentives or packages also decreased teacher morale.

### 6.3. Retention and Attrition Intentions

#### 6.3.1. Theme 8: White-Collar Jobs

Some of the teachers were prepared to abandon their posts in pursuit of white-collar jobs within or outside the country as a result of poor working conditions, which were exacerbated by the COVID-19 pandemic. They considered other professions to be earning more income than themselves. Moreover, they perceived some professions such as the health and business sector to be more respectable than the teaching profession.

> "Because teachers' salary is too small, people don't respect teachers. I don't have plans to retire from this profession. Even your pension payments will be small. As I said earlier, the job is very tedious and the salary is meagre. It is too burdensome. A lot of hardship. Your salary will not even meet your demands. So I wish to enter into a different profession if I get the opportunity"—John

> "Yes, and no, If I find any other lucrative job which is better than the teaching profession, I will quit. But if I don't get I will also remain in it till retirement. That is why I said yes and no"—Joyce

#### 6.3.2. Theme 9: Further studies

> "Yeah, if I get the chance I would like to top up. You are more respected and get more leadership roles when your qualifications are high. Although the working situation is poor, some people in higher positions are relatively okay. So, I plan to top up one day"—Martin

#### 6.3.3. Theme 10: Promotion

Some of the teachers revealed that they planned to remain in the profession because being a veteran in teaching can facilitate the promotion process. That is, because of promotion opportunities, some of the participants in the study were ready to endure any hardship such as those associated with COVID-19.

> Higher qualifications and teaching experience can ensure that you have higher chances of promotion. So, I intend to stay and get promoted. I've already worked for close to fifteen years, so why not stay and get promoted? As for the hardship, we are used to it . . . yes, the COVID-19 affected all of us but for me, I am used to such problems"—Bright

Low teacher work motivation can result in brain drain. As can be deduced from the participants' responses, many teachers in the profession had the intention to leave the profession. Although in Ghana, new teachers are recruited annually after graduation, the attrition of teachers already in the profession is quite alarming and needs to be addressed [27,30]. In times of disaster such as COVID-19, issues negatively affecting the motivation of teachers need to be addressed to prevent any pandemic-induced teacher attrition [37].

### 6.4. Navigating Pedagogical Pathways

#### 6.4.1. Theme 11: Intrinsic motives

The inherent desire to teach is why most of the participants were not discouraged by the working conditions that were worsened by COVID-19. Most participants in the study expressed a passion for their job and their innate desire to help others.

"Ok. I saw that it is an alternative way to render my service to people. That is why I chose to be a teacher in Ghana. I had wanted to be in the health sector to also help people. What keeps me motivated to teach every day is that intrinsically, I have decided to use teaching as an alternative way to render my service to people. So, what motivates me is to assist people and help people acquire knowledge. So even during the COVID-19, I was prepared to teach wholeheartedly"—Belinda

"What keeps me motivated to teach every day is inner passion. My inner passion can keep me in the profession for a long time or till retirement. Sometimes too you meet lovely kids, so for the sake of the students. I also chose to be a teacher because it is a noble job"—Gifty

6.4.2. Theme 12: Altruistic motives

Some of the teachers also felt an obligation to society and wanted to give back through their work. They considered teaching as a good avenue to contribute to society and student development.

"Why I teach social studies is, teaching people to understand their citizenship, the needs and the responsibilities of society makes me feel the teaching of the subject is important. Yeah, so when I teach and make people feel that they are responsible for their own behaviour and everything concerning the environment, for them to understand the pros and cons of their actions affecting society... If they're able to understand it makes me very happy . . . I used the opportunity to talk about the need to protect the body against infection during the pandemic"—Robert

Rather than "surviving", teachers who are intrinsically and altruistically motivated are able to "thrive", even in distressing times such as COVID-19 [14]. Teachers with intrinsic–altruistic motives are able to endure hardship and persevere in difficult circumstances to deliver quality instruction. They are self-determined to help students [56,57].

## 7. Implications

It is obvious from the study's findings that COVID-19 altered the lifestyle of teachers and negatively affected their work motivation. The large-scale disruption in education systems was strongly felt by educational institutions in developing countries. Teachers in such institutions suffered from an increased level of stress due to the unprecedented nature of the challenges that ensued after the pandemic. In this study, some of the stressors that decreased the work motivation of teachers were the lack of adequate infrastructure to facilitate learning, the deficient digital skills of teachers for online instruction, financial difficulties and low remuneration, inadequate teaching materials, and difficulties associated with the "double-track" system. UNESCO [93] advocates for support for teachers who are frontline workers in the education sector and have demonstrated a high level of commitment in these crisis times. Setting up measures and support systems that enhance their work motivation is one way of recognizing their efforts in response to the health crisis and disruption in education as a result of the COVID-19 pandemic. That is, teachers should not be left out in the prioritization of the well-being and safety of students and other educators.

Some of the teachers in the profession possessed the resilience to instruct students in these challenging times because of the intrinsic and altruistic value of teaching. However, external factors such as the COVID-19 crisis have the potential to undermine their innate desire to help students and contribute to social growth and development [1,56,57,94]. All contextual factors that threaten the inherent desire of teachers to promote progressive education and lifelong learning should be removed.

Chief among the concerns to be addressed is the psychological consequences of the pandemic on teachers. When teachers are emotionally distressed, they produce a subnormal performance, which ultimately affects the success of students. As has been reviewed earlier, depressive symptoms such as sadness, loneliness, paranoia, and suicidal

ideation can develop in teachers because of the negative impact of the COVID-19 crisis on their mental health. School leaders have to ensure that psychological units in their schools are functioning to render quality mental health support for teachers.

Some of the teachers in the current study also had attrition intentions after perceiving a lack of governmental and institutional support in the wake of the pandemic. Teacher attrition is a great problem facing the educational sector in many countries. To avoid brain drain and labour shortage as far as the teaching field is concerned, it is vital for school authorities to lay out intervention strategies that keep quality teachers in the profession. For example, the stigma and discrimination of COVID-19 experienced by teachers need to be addressed. Additionally, professional networks should be created for teachers for them to support one another in terms of training. During the COVID-19-inspired online learning, teachers with technical skills could have coached their peers on digital instruction.

Theoretically, this study calls on researchers and practitioners to find ways to build a crisis-resistant framework for sustainable quality education. By addressing factors that serve as a barrier to the work motivation and job satisfaction of teachers, educators would be able to develop a workforce who are ready to persevere in difficult times such as this COVID-19 era to ensure school success. Timely remuneration, recognition for school frontliners, professional training, and good school resources are some of the ways to boost the morale of teachers for optimal learning.

## 8. Conclusions

This qualitative inquiry took a comprehensive and fresh look at teacher work motivation in light of the COVID-19 pandemic. Thirty (30) core subject teachers from "Category B" secondary schools in the country were recruited to provide answers to the research questions. Findings suggest that COVID-19 gravely affected the work motivation of teachers. That is, generally, it can be surmised that teachers suffered from low work motivation due to negative extrinsic factors of COVID-19. Teachers received both government and institutional support, but their experiences in the reception of support were at times negative. For example, some of the teachers mentioned that they did not receive any special motivational package, although they were aware that the government and their institution provided incentives for some colleagues. Schools also had limited funds for operations.

Teachers also suffered from burnout as a result of increased workload and pressure associated with instruction during the pandemic. An example is the prolonged teaching and limited holidays after the implementation of the "double-track" system. Teachers had limited time to prepare for the separate group of teachers due to the nature of the academic calendar. All these factors occurred in tandem with caring for the emotional and safety needs of students. Consequently, it was identified that some of the teachers had intentions to leave the profession. Nonetheless, traits such as passion and the innate desire to promote student development and contribute to society were some of the reasons why teachers in the study persevered in teaching amid the COVID-19 crisis. Nonetheless, negative extrinsic factors can undermine intrinsic–altruistic motives, thereby worsening the attrition rate and turnover intentions of teachers. It is recommended for policymakers in education and educators to build upon the intrinsic and altruistic motives of teachers, and for them to place a higher value on teaching.

Future studies can adopt a sophisticated technique such as mixed methods to add a quantitative tone to investigating the topic. The key measures needed to mitigate the negative experiences of teachers and how they should be implemented can be spotlighted in the new study. A limitation of this study is that only the perspectives of core subjects were solicited and teachers in the study came from only one geographical region in the country. In a quantitative study, more teachers can be recruited and those in a different category of school can be used. The generation of codes was also performed by only the researcher and one independent reviewer. More reviewers could have strengthened the trustworthiness and validity of the findings of the study.

**Funding:** This research received no external funding.

**Institutional Review Board Statement:** The study was conducted according to the guidelines of the Declaration of Helsinki, and approved by the Ashanti Region Education Office, Ghana.

**Informed Consent Statement:** Informed consent was obtained from all subjects involved in the study.

**Data Availability Statement:** The data presented in this study are available on request from the corresponding author.

**Conflicts of Interest:** The authors declare no conflict of interest.

## Appendix A. Interview Guide

Topic: Researching Teacher Work Motivation in Ghana through the lens of COVID-19
Date:
Time interview started:
Time interview ended:

*Part I-Personal Information*

Age:
Gender:
Subject taught:
Teaching experience:
Drafted interview questions

1. How do you perceive online instruction during COVID-19?"
2. What major challenges did you face during the instruction?
3. How did you perceive your professional identity during the pandemic teaching?
4. Share with me some of your experiences during the face-to-face instruction.
5. Probe: How do you perceive the double-track system and its effect on your teaching?
6. Tell me some of the main psychological stressors you experienced with teaching.
7. What kind of school resources was available to assist you in your job during the pandemic?
8. What were the main support systems for teachers during the pandemic?
9. Probe: Please, talk about both government support and school initiatives to help your well-being and teaching.
10. What impact has COVID-19 and its after-effects had on your turnover and attrition intentions?
11. How you do navigate through the challenges?
12. What are the main reasons why you endured the challenges and instructed students?

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
