# Peer review of "Researching Teacher Work Motivation in Ghana through the Lens of COVID-19"

_covid, doi:10.3390/covid3020023_

Round 1

Reviewer 1 Report

In the background, it is necessary to add a little data (possibly the results of interviews) with several teachers, regarding motivation and resilience in teaching during and before the pandemic occurred

The results are quite clear, but the discussion needs to be sharpened a little with the researcher's review and efforts should be made to refer to previous research.

The conclusion should contain the researcher's resume and conclusions regarding the research question. Need to explain a little about the conclusion. For example: how far is the teacher's motivation --> then the researcher must present a conclusion about "to what extent".

Regarding questions 2, 3 and 4 in the conclusion it is necessary to explain a little. Don't just say yes or no results. For example: the teacher's response to government assistance is negative --> describe a little "what are the negatives".

Some referrals could not be opened/found). For example number (1), (2), (5)

Author Response

Reviewer 1

  1. In the background, it is necessary to add a little data (possibly the results of interviews) with several teachers, regarding motivation and resilience in teaching during and before the pandemic occurred

Author’s Response:

Thanks very much! Recent studies on the topic have been assessed and interview data or research findings relating to the work motivation and resilience of teachers before and during the pandemic have been included in the background section.

  1. The results are quite clear, but the discussion needs to be sharpened a little with the researcher's review and efforts should be made to refer to previous research.

Author’s response:

Thank you. The discussion is embedded in the findings section and relevant and recent literature has been cited. Additionally, based on the comment, a few explanatory sentences have been included in the implication section.

  1. The conclusion should contain the researcher's resume and conclusions regarding the research question. Need to explain a little about the conclusion. For example: how far is the teacher's motivation --> then the researcher must present a conclusion about "to what extent".

Author’s response:

Thanks! It is included the in the conclusion that teachers suffered from a low work motivation.

  1. Regarding questions 2, 3 and 4 in the conclusion it is necessary to explain a little. Don't just say yes or no results. For example: the teacher's response to government assistance is negative --> describe a little "what are the negatives".

Author’s response:

Thanks! Examples are given to support some findings summarised in the conclusion section.

  1. Some referrals could not be opened/found). For example number (1), (2), (5)

Author’s response: Thanks. This point was not particularly clear but all references in the study can be accessed. The mentioned references were searched and were found.

Reviewer 2 Report

This paper describes a qualitative phenomenological study involving thirty (30) secondary school teachers in Ghana who took a comprehensive and fresh look at how COVID-19 impacted the work motivation of teachers. It was found that teachers suffered a great deal of stress in the wake of the pandemic and had face-mounting concerns about their working conditions. 

The paper must be revised for the related works. It might be good to include a table with entries linked to similar works with citations. Also few plots must be included based on the research data acquired to give an impact to readers about the research results.

Author Response

  1. This paper describes a qualitative phenomenological study involving thirty (30) secondary school teachers in Ghana who took a comprehensive and fresh look at how COVID-19 impacted the work motivation of teachers. It was found that teachers suffered a great deal of stress in the wake of the pandemic and had face-mounting concerns about their working conditions. 

The paper must be revised for the related works. It might be good to include a table with entries linked to similar works with citations. Also few plots must be included based on the research data acquired to give an impact to readers about the research results.

Author’s response:

Thank you very much! The study has been revised and a few recent literature has been added. Because the study did not adopt a quantitative design and did not calculate the frequencies of codes generated, graphical representations were not needed. The themes have been summarized in a table form for easy access and readership.

Reviewer 3 Report

The topic is addressing a critical issue during the difficult time of Covid 19.

The report mentioned on page 3 line 99 and onwards by secretary of GNAT related to the attrition rate of teachers seems to contradict what the study resulted in finding teachers motivated to continue their work despite the difficult situation in schools and deteriorated faculties and internet infrastructure. Is there any reason why this is the case?

Methodology is appropriate and details of the participants are clearly stated.

Suggest to include the interview questions to the study and also to unpack what type of online activities were used during the sessions in comparison to the type of activities in face to face modalities.

Further details to describing the interview questions are needed and could be added to the text on page 7 line 323.

Also reference to the details of the studies used to develop the interview question would help (page 7, line 324/5)

More details to explain further how the coding was made page 7, line 346 and whether more researchers were included in this analysis to avoid subjectivity.  Also include this point in the study limitations.

On page 8, 364 clearly state the exact number of teachers rather than just mentioning ‘some’.

Table 1 is very helpful and shows clearly the themes and their categories. Suggest to add a column that relates these themes to the RQs.

Author Response

  1. The topic is addressing a critical issue during the difficult time of Covid 19.

The report mentioned on page 3 line 99 and onwards by secretary of GNAT related to the attrition rate of teachers seems to contradict what the study resulted in finding teachers motivated to continue their work despite the difficult situation in schools and deteriorated faculties and internet infrastructure. Is there any reason why this is the case?

Author’s response:

Thank you! The information about teacher attrition as reported by GNAT is used to give a background context of the rate of attrition in Ghana over a few years. The current study’s findings also found that teachers had attrition intentions. However, teachers were probed further on why they were still teaching at the time and how they were able to overcome or deal with the emerging challenges. About six of the teachers indicated that the passion they had for teaching is the reason why they remained in the profession and instructed students during the pandemic period. At the same time, it is mentioned that negative extrinsic factors can undermine teachers’ intrinsic-altruistic motives such as passion. Hence, a call is made for educators and policymakers to remove any obstacle that curtails or diminishes the inherent desire for teachers to teach.

  1. Methodology is appropriate and details of the participants are clearly stated.

Suggest to include the interview questions to the study and also to unpack what type of online activities were used during the sessions in comparison to the type of activities in face to face modalities.

Author’s response:

Thank you! A sample of the interview questions has been included in the method section, specifically, the “instrument” section. The major online activities during the pandemic in secondary schools have also been included under the first theme. They were virtual platforms, radio and television programs. Additionally, ICT centres in schools were employed by some teachers for online instruction.

  1. Further details to describing the interview questions are needed and could be added to the text on page 7 line 323.

Also reference to the details of the studies used to develop the interview question would help (page 7, line 324/5)

Author’s response:

How the interview questions were developed and relevant studies that guided the formulation of questions have been incorporated in the study.

  1. More details to explain further how the coding was made page 7, line 346 and whether more researchers were included in this analysis to avoid subjectivity.  Also include this point in the study limitations.

Author’s response:

Thank you! The information has been added. Codes were manually generated by the researcher. It was reviewed severally by the researcher and finally by an independent researcher with insight into the topic and context.

  1. On page 8, 364 clearly state the exact number of teachers rather than just mentioning ‘some’.

Author’s response:

Thank you! This has been corrected.

  1. Table 1 is very helpful and shows clearly the themes and their categories. Suggest to add a column that relates these themes to the RQs.

Author’s response:

Thank you! This has been added.

Special thanks to all the reviewers for their insightful feedback.

Round 2

Reviewer 2 Report

This paper describes the vulnerability assessment of teachers during the COVID-19 pandemic.  Due to the COVID-19 crisis, the negative experiences of teachers in their work environments ere found to augment during both online and physical instruction. This study involves thirty (30) secondary school teachers in Ghana to assess the impact of the work motivation of teachers. It was found that teachers suffered a great deal of stress in the wake of the pandemic and had face-mounting concerns about their working conditions. 

It is important to represent data analysis results in graphical and/or tabular form. Also, if the assessment is made based on a questionnaire, it should also be included in the paper. Otherwise, these claims can not be proved.

Author Response

Thanks very much! Please, Table 1 has the themes and sub-themes that were generated from the codes. It has been highlighted in yellow. The interview guide used in soliciting answers from the participants has also been added to the revised version of the manuscript. Demographic details about participants are included in the method section; "Participants and sampling." Also, in revising the manuscript, grammatical errors and minor language issues have been fixed. Thank you!